# The Effectiveness of Extruded-Cooked Lentil Flour in Preparing a Gluten-Free Pizza with Improved Nutritional Features and a Good Sensory Quality

**DOI:** 10.3390/foods11030482

**Published:** 2022-02-07

**Authors:** Antonella Pasqualone, Michela Costantini, Michele Faccia, Graziana Difonzo, Francesco Caponio, Carmine Summo

**Affiliations:** Department of Soil, Plant and Food Science (DISSPA), University of Bari Aldo Moro, Via Amendola, 165/A, I-70126 Bari, Italy; michela.costantini.92@gmail.com (M.C.); michele.faccia@uniba.it (M.F.); graziana.difonzo@uniba.it (G.D.); francesco.caponio@uniba.it (F.C.); carmine.summo@uniba.it (C.S.)

**Keywords:** legumes, extrusion-cooking, pizza, hydrocolloid, starch gelatinization, pasting properties, Mixolab, viscoamylograph, bioactive compounds, consumer liking

## Abstract

Extruded-cooked lentil (ECL) flour was used to fortify (10/100 g dough) gluten-free pizza, which was compared with rice/corn-based pizza (control), and with pizza containing native lentil (NL) flour. Viscoamylograph and Mixolab data evidenced the hydrocolloid properties of ECL flour (initial viscosity = 69.3 BU), which contained pregelatinized starch. The use of ECL flour made it possible to eliminate hydroxymethylcellulose (E464), obtaining a clean label product. Both NL and ECL pizzas showed significantly (*p* < 0.05) higher contents of proteins (7.4 and 7.3/100 g, respectively) than the control pizza (4.4/100 g) and could be labelled as “source of proteins” according to the Regulation (EC) No. 1924/2006. In addition, NL and ECL pizzas were characterized by higher contents of bioactive compounds, including anthocyanins, and by higher in vitro antioxidant activity (1.42 and 1.35 µmol Trolox/g d.m., respectively) than the control pizza (1.07 µmol Trolox/g d.m.). However, NL and ECL pizzas also contained small amounts of undigestible oligosaccharides, typically present in lentils (verbascose = 0.92–0.98 mg/g d.m.; stachyose = 4.04–5.55 mg/g d.m.; and raffinose = 1.98–2.05 mg/g d.m.). No significant differences were observed in the liking level expressed by consumers between ECL and control pizzas.

## 1. Introduction

In recent years, the market of gluten-free (GF) products has increased because these products are in demand, not only by celiac patients, but also by non-celiac consumers [1]. The latter look for a GF diet due to symptoms triggered by the ingestion of gluten (non-celiac gluten sensitivity) [2,3], or because they perceive this diet as a way to control weight and be healthy [4,5]. However, the nutritional composition of GF foods may be worse than their gluten-containing counterparts. A frequent problem is the low amount of minerals and vitamins, coupled with a high amount of saturated fat and salt, raising concerns about the long-term effects on health [6,7,8]. Therefore, there is a need to improve the nutritional quality of GF foods.

Legumes are a sustainable crop rich in proteins, fibers, and micronutrients [9]. Many studies have demonstrated the beneficial effects of legumes on health, resulting in a lower incidence of coronary heart disease, colon cancer, diabetes mellitus, hypertension, and gastrointestinal disorders [10,11,12]. In recent years, numerous studies have focused on the innovative food applications of legumes. Legume-based dairy-free cheese alternatives, as well as burgers, salad dressings, pasta, and bakery products have been proposed [10,13,14,15,16,17,18]. Moreover, pregelatinized legume flours, prepared by means of extrusion-cooking, show hydrocolloid properties, making them suitable for the formulation of GF baked goods [19,20,21]. After rehydration, in fact, the pregelatinized starchy fraction of extruded-cooked legume flours forms a viscous medium, allowing the entrapment of air bubbles in the GF dough [22]. The lentil is one of the most cultivated leguminous food crops, with a world production of about 6 million tons in 2019 [23]. Several studies have evidenced that extrusion-cooking improves the nutritional and techno-functional properties of lentil flour [24,25,26].

Pizza is an ancient food with a rich history and a strong cultural significance: in 2010, the Neapolitan-style pizza received the “Traditional Specialty Guaranteed” mark at the European level [27]; in 2017, the “art of Neapolitan pizza makers” was awarded the “Intangible Cultural Heritage of Humanity” recognition by the United Nations Educational, Scientific and Cultural Organization (UNESCO) [28]. Originating in Italy but appreciated worldwide, pizza is one of the most popular convenience food products. Many GF versions of pizza are currently marketed, but their nutritional quality is often questionable. A survey on the nutritional composition of GF products, mostly consumed in Spain, evidenced twice as much total fat, mainly saturated, in the GF dough/pastry/pizza products than in their equivalents with gluten [29]. Moreover, in many cases the commercial GF pizza is closer to a cake than a conventional wheat-based pizza, which requires a sheetable dough. Therefore, the sensory quality of GF pizza is often low, due to its crumbling texture and poor mouthfeel, flavor, and color [30]. The quality defects of pizza and other GF baked goods may contribute, among other factors, to the long-term lack of adherence to a strict GF diet, which is relatively frequent in adults [31]. A single study has focused on the sensory quality improvement of GF pizza so far, by combining a variety of starches and proteins with a microencapsulated high-fat powder [30]. This study, however, did not take into account the nutritional implications of the proposed approach.

In a previous study, the fortification with chickpea flour was found to be effective in improving the nutritional profile of conventional (gluten-containing) pizza without worsening its quality [17]. No study, however, has considered the effect of the addition of legume flour in the preparation of GF pizza. Therefore, the aim of this research was to evaluate the effectiveness of extruded-cooked lentil flour to improve the nutritional profile of GF pizza, maintaining a good sensory quality. The experimental pizza, containing extruded-cooked lentil flour, was compared with rice/corn-based GF pizza and with GF pizza enriched with non-extruded lentil flour.

## 2. Materials and Methods

### 2.1. Basic Ingredients

Rice flour (Bongiovanni, Villanova Mondovì, Italy), corn flour (EcorNaturasì SpA, Verona, Italy), corn starch (Bongiovanni, Villanova Mondovì, Italy), baker’s yeast (*S**accharomyces cerevisiae*, Lievitalia SpA, Parma, Italy), and sea salt (Com-Sal srl, Pesaro, Italy) were purchased from local retailers. Hydroxypropylmethylcellulose (HPMC, E464) was provided by Laboratori Bio Line (Canaro, Italy). Psyllium (*Plantago ovata* Forsk) seed husk powder was provided by Erbavoglio (Brescia, Italy).

### 2.2. Preparation of Native and Extruded Lentil Flours

The production of native lentil (NL) flour and extruded-cooked lentil (ECL) flour was made at a local company (Andriani SpA, Gravina in Puglia, Italy), which provided the raw material, i.e., dehulled lentils (*Lens culinaris* Medik.). Lentils were divided into two batches, submitted to the processing steps schematized in Figure 1. In detail, the first batch was milled using an industrial hammer mill (DNZF-0655 Fine Grinding Mill, Bühler, Uzwil, Switzerland) equipped with a screen that had 3 mm holes. The obtained flour was then conditioned to 28 ± 1 g 100 g^−1^ moisture and was processed by an industrial co-rotating twin-screw extruder-cooker (PRIOtwin-BCTF, Bühler, Uzwil, Switzerland) at a 500 kg/h feed rate (industrial conditions) with a 3:1 compression ratio, a 220-rpm screw rotation speed, and 100 °C die head temperature. The extrusion-cooking conditions were selected from previous research based on the cold viscosity of the extruded-cooked flour obtained, which had to be acceptably high (69.3 Brabender units—BU) [26]. The obtained pellets, extruded through a die with 4 mm holes, were dried in an industrial drier (Aeroglide, Bühler, Uzwil, Switzerland) at 110 °C for 14 min, then at 115 °C for 16 min, to reach 12 g 100 g^−1^ moisture. The dried pellets were then milled using an industrial four-roller mill (MDDP, Bühler, Uzwil, Switzerland) consisting of four break rolls (B1A, B1B, B2, and B3), and four reduction rolls (C1A, C1B, C2, and C3), followed by a plansichter (MDPK, Bühler, Uzwil, Switzerland) with 0.2 mm hole sieves, to produce the ECL flour. The second batch of dehulled lentils was directly roller-milled in the same milling conditions (an industrial four-roller mill and a plansichter with 0.2 mm holes) to produce the NL flour.

### 2.3. Preparation of Pizza Crust

Three types of pizza crusts were produced at Panificio Antico Forno (Bitonto, Italy) according to the formulations reported in Table 1, with control pizza, pizza with NL flour, and pizza with ECL flour. NL and ECL partly substituted the rice flour at a level of 10/100 g dough. The optimal water amount was assessed in the preliminary trials. A straight dough process was performed, as follows. Flours (rice, corn, NL, and ECL), corn starch, psyllium seed husk powder, HPMC, and salt were mixed at low speeds (around 100 rpm) for 2 min by a spiral kneader (Sprint 30, C.M. Sottoriva, Vicenza, Italy). Meanwhile, baker’s yeast was dissolved in the total amount of water and was then added to the solid ingredients, followed by kneading for 10 min at high speed (around 200 rpm). The homogeneous dough obtained was divided into 220 g portions, which were manually rolled out with a rolling pin to obtain circular discs (about 30 cm in diameter, 4 mm thick). The dough discs were then proofed for 1.5 h at 30 °C with a relative humidity (RH) = 70% (EKL 1264 proofer, Tecnoeka S.r.l., Borgoricco, Italy), pierced by a roller with thin punches, and baked in an electric oven (SudForni srl, Casoria, Italy) at 220 °C for 18 min.

### 2.4. Determination of Nutritional Composition

Protein (N × 5.7) content was determined, as described in the American Association of Cereal Chemists (AACC) method 46–11.02 [32], by using a DK 6 Kjeldahl digestion unit (Velp Scientifica srl, Usmate, Italy) and a UDK 126 Kjeldahl distillation unit (Velp Scientifica srl, Usmate, Italy). Kjeldahl tablets (a catalyst with 1.5% CuSO_4_ · 5H_2_O and 2% Se), H_2_SO_4_, NaOH, and B(OH)_3_ were all from Sigma-Aldrich Chemical Co. (St. Louis, MO, USA). The moisture content was determined at 105 °C by means of an automatic moisture analyzer (Radwag Wagi Elektroniczne, Radom, Poland) according to the AACC method 08–01 [32]. Fat was extracted and determined by the Soxhlet method [33], with a diethyl ether as the solvent, by using a SER 148 semi-automatic extraction system (Velp Scientifica srl, Usmate, Italy). The total dietary fiber was determined by the enzymatic–gravimetric procedure, according to the AOAC method 991.43 [34], by using a total dietary fiber analyzer composed of a thermostatic incubator and a filtration unit (mod. Csf6, Velp Scientifica srl, Usmate, Italy). Alfa-amylase, protease, amyloglucosidase, 2-morpholinoethanesulfonic acid (MES), and tris(hydroxymethyl)aminomethane (TRIS), used for the total dietary fiber determination, were all from Sigma-Aldrich Chemical Co. (St. Louis, MO, USA). Carbohydrates were calculated by difference. The energy value (kJ) was calculated using Atwater general conversion factors by considering the contribution of 2 kcal/g from the total dietary fiber, according to Annex XIV of the Regulation (EC) No. 1169/2011 [35].

All analyses were carried out in triplicate.

### 2.5. Determination of Oligosaccharides

Oligosaccharides (verbascose, stachyose, and raffinose) were determined by high-performance liquid chromatography (HPLC) (Agilent Technologies, Santa Clara, CA, USA), equipped with a 300 × 7.8 mm cation exchange column (Rezex RCM column, Ca^2+^, 8 μm, Torrance, CA, USA) and a refractive index detector (RID 1260, Agilent Technologies, Santa Clara, CA, USA), as follows: Each sample (10 mg) was suspended in 10 mL of deionized water and, after 5 min of stirring, the suspension was filtered through a 0.22 µm cellulose acetate filter. The HPLC separation was conducted isocratically at a flow rate of 0.8 mL/min, a column temperature of 80 °C, and a RID temperature of 40 °C. The deionized water was used as the mobile phase. The identification was carried out by comparing the retention time with that of the corresponding standard (Meck KGaA, Darmstadt, Germany). A calibration curve for each oligosaccharide was prepared for the quantification. The analysis was carried out in triplicate.

### 2.6. Determination of Bioactive Compounds and Antioxidant Activity

The total anthocyanins were quantified by adding 0.1 g of each sample to 1 mL of 85:15 (*v*/*v*) methanol/1 M HCl, stirring consistently for 30 min in the dark, then centrifuging for 5 min at 12,000× *g* and measuring the absorbance of the clear extracts at 535 nm by a Cary 60 UV–Vis spectrophotometer (Agilent Technologies Inc., Santa Clara, CA, USA) after setting a calibration curve with cyanidin 3-*O*-glucoside (PhytoLab, Vestenbergsgreuth, Germany) as the standard [36].

The total phenolic compounds were determined as reported in [37]. In detail, 0.1 g of each sample were added of 1 mL of methanol and were stirred consistently for 2 h. Then, the suspension was centrifuged at 10,000× *g* for 5 min and the recovered supernatant (100 μL) was added, with 500 μL of the Folin–Ciocalteu reagent (Sigma-Aldrich Chemical Co., St. Louis, MO, USA) and 2 mL of 15% (*w*/*v*) sodium carbonate. The final volume brought up to 10 mL with distilled water. After 1 h in the dark, and centrifugation at 12,000× *g* for 3 min to precipitate any particles, the absorbance of the solution was measured at 765 nm by a Cary 60 UV–Vis spectrophotometer (Agilent Technologies Inc., Santa Clara, CA, USA) after setting the calibration curve with the methanol solutions of ferulic acid (Sigma-Aldrich Chemical Co., St. Louis, MO, USA) as the standard.

Total carotenoid pigments were determined by suspending 1 g of each sample in 5 mL of water-saturated *n*-butyl alcohol in an orbital shaker for 3 h at 260 rpm, then centrifuging for 5 min at 12,000× *g* and measuring the absorbance of the clear extracts at 435.8 nm using a Cary 60 UV–Vis spectrophotometer (Agilent Technologies Inc., Santa Clara, CA, USA). The total carotenoid content was expressed as beta-carotene, and calculations were made based on the extinction coefficient of 1.6632 for a solution of 1 mg of beta-carotene in 100 mL of water-saturated *n*-butyl alcohol, according to the AACC method 14–50.01 [32].

The antioxidant activity was assessed by a 2,2′-azino-bis(3-ethylbenzothiazoline-6-sulphonic acid) (ABTS) (Sigma-Aldrich Co., Milan, Italy) test, as reported in [38]. The test is based on the capacity of a sample to inhibit the ABTS radical (ABTS•+) compared with a reference antioxidant standard (6-hydroxy-2,5,7,8-tetramethylchroman-2-carboxylic acid, Trolox) (Sigma-Aldrich Chemical Co., St. Louis, MO, USA). In detail, 0.1 g of each sample was added to 1 mL of methanol in an orbital shaker for 2 h, in the dark, then centrifuging at 10,000× *g* for 5 min. In parallel, ABTS•+ was generated by mixing 25 mL of 7 mM ABTS and 440 μL of a 2.45 mM potassium persulphate (K_2_S_2_O_8_) aqueous solution, allowing the solution to stand for 16 h in the dark at room temperature. The solution containing ABTS•+ was then diluted with water to obtain an absorbance of 0.80 ± 0.1 at 734 nm, measured by a Cary 60 UV–Vis spectrophotometer (Agilent Technologies Inc., Santa Clara, CA, USA). Each sample extract (100 μL) was then added to 3.9 mL of diluted ABTS•+solution, and after 5 min the absorbance at 734 nm was measured. Solutions of Trolox (Sigma-Aldrich Chemical Co., St. Louis, MO, USA) were prepared at a concentration ranging from 20 to 1000 μM (*R*^2^ = 0.9965) for the calibration.

All analyses were carried out in triplicate.

### 2.7. Viscoamylograph Analysis

Gelation properties of flours were determined using a viscoamylograph (Brabender Instruments, Duisburg, Germany) as reported in [26]. The viscosity values at the beginning (initial viscosity, IV), at the peak (peak viscosity, PV), at the end of the holding period at 95 °C (minimum viscosity, MV), and at the end of the cooling phase (cooling maximum viscosity, CMV) were recorded and expressed in arbitrary Brabender units (BU). The differences between PV and MV, and between CMV and MV, were defined as a “breakdown” and a “setback”, respectively. The analysis was conducted in triplicate.

### 2.8. Mixolab Analysis

The gelation properties and dough mixing properties of NL and ECL flours were determined using the Mixolab instrument (Chopin Technologies, Villeneuve-La-Garenne, France) in the conditions described in [26]. The maximum torque produced by dough during mixing (C1), and the minimum torque while the dough was subjected to mechanical and thermal stress (C2), were recorded. The analysis was conducted in triplicate.

### 2.9. Physical Determinations

The pliability of the pizza samples was evaluated by using a Z1.0 TN texture analyzer (Zwick Roell, Ulm, Germany) equipped with a 1 KN load cell. Data were acquired by means of the TestXPertII v. 3.41 software (Zwick Roell, Ulm, Germany). The three-point bending test was performed, as described in [39], with minimal modifications. A 50 × 20 mm strip, cut from the central zone of the pizza samples, was horizontally placed upon two vertical bars spaced 40 mm apart. After a preload of 5 g at a speed of 5 mm/s, the strip was bent by a third bar that descended for 12 mm at the speed of 1 mm/s. The F_max_ (maximum force applied to the sample, expressed in N/mm^2^) was recorded. The analyses were carried out in triplicate.

The diameter (D) and thickness (T) of the pizza crust before and after baking were determined by a caliper and were used to calculate the percentage of variation due to baking, as follows: % of variation of D (or T) = [D (or T) after baking − D (or T) before baking]/D (or T) before baking × 100.

The analyses were carried out in triplicate.

The color indices *L**, *a**, and *b** were measured by using a Chromameter CM-600d (Konica Minolta, Tokyo, Japan) with a D65 illuminant and a CIE 10 standard observer angle. Five replicated analyses were made. The total colour variation (∆E) was then calculated to compare the differences between the control pizza and the two types of lentil-enriched pizza, according to the following equation:$$\Delta E=\left[{\left(\Delta L^{*}\right)}^{2}+{\left(\Delta a^{*}\right)}^{2}+{\left(\Delta b^{*}\right)}^{2}\;\right]{\;}^{1/2}$$

The following scale was considered: ΔE = 0–0.5, which indicated a very low difference; 0.5–1.5, which indicated a slight difference; 1.5–3.0, which indicated a noticeable difference; 3.0–6.0, which indicated an appreciable difference; 6.0–12.0, which indicated a large difference; and >12.0, which indicated a very obvious difference [40].

### 2.10. Sensory Analysis

A quantitative descriptive analysis (QDA) was performed, according to the International Standardization Organization (ISO) standard 13299 [41], by a sensory panel consisting of eight trained members. The panelists (four male and four female, with an age range from 37 to 54 y), experienced in the sensory evaluation of cereal-based foods [42], had neither food allergies nor intolerances and were regular consumers of pizza and lentils. The reason for recruiting non-celiac panelists was because the production trials could not guarantee a GF environment.

Pre-test sessions were carried out: (i) to define the list of descriptors to be evaluated; (ii) to define the intensity range of each descriptor; (iii) to fix the scale anchors of each descriptor; and (iv) to verify the reliability, consistency, and discriminating abilities of the panelists, according to the ISO standard 11132 [43]. The study protocol followed the ethical guidelines of the laboratory. Panelists were given information about the study aims, and individual written informed consent was obtained from each participant.

Samples were presented in dishes coded with three-digit random numbers and were distributed simultaneously in a random order. Sensory evaluations took place in a conference room, where temporary partitions were used to set up isolated tasting booths to separate panelists during the analysis, in agreement with the ISO standard 8589 [44]. Testing was performed at ambient room temperature (20 ± 2 °C).

A total number of eight sensory descriptors were considered. In the order of evaluation, they were: olfactory descriptors (global odor and lentil odor), visual–tactile descriptors (color, consistency, and pliability), taste descriptors (sweetness and saltiness) and flavor descriptors (lentil flavor). The descriptors were rated on a structured scale that provided a 0–9 score range (contractual units—c.u.). The scale anchors for global odor, lentil odor, sweetness, saltiness, and lentil flavor were: 0 c.u. = minimum intensity; 9 c.u. = maximum intensity. The scale anchors for color were: 0 c.u. = ivory; 9 c.u. = carrot orange with rust undertones. The scale anchors for consistency (to be evaluated by compressing pizza samples with fingers) were: 0 c.u. = very soft and tender, like a brioche; 9 c.u. = hard, not very compressible. The scale anchors for pliability were: 0 c.u. = inflexible, rigid, and with a tendency to break; 9 c.u. = very pliable and elastic, easily bending without breaking. The sensory analysis was carried out in triplicate.

### 2.11. Consumer Tests

Seventy consumers (37 females and 33 males aged between 30 and 65) that were not affected by food allergies or intolerances, and were regular consumers of pizza and lentils, evaluated the pizza samples. They were given information about study aims, and individual written informed consent was obtained from each participant. The consumers were asked to state if they liked or disliked each pizza sample, rating their disliking or liking levels on a structured scale from 1 to 5 c.u. (1 c.u. = disliked a little or liked a little; 5 c.u. = disliked a lot or liked a lot).

### 2.12. Statistical Analyses

Statistical analyses were carried out using the Minitab 17 Statistical Software (Minitab, Inc., State College, PA, USA, 2010). Significant differences were determined at *p* < 0.05 by a one-way analysis of variance (ANOVA) followed by Tukey’s HSD test. The results of the consumers test were statistically analyzed by the Friedman test using XLStat software (Addinsoft SARL, New York, NY, USA).

## 3. Results and Discussion

### 3.1. Characteristics of Flours and Criteria for Pizza Formulation

Pizza formulation is reported in Table 1, while the nutritional characteristics of the flours used in the experimental trials are reported in Table 2.

In setting up the formulation of the control pizza, corn and rice flours were used in a 1:4 ratio. The quantity of corn flour, typically bright yellow, was smaller than rice to obtain, in the mixture, a yellowish color similar to that of a conventional wheat-based dough. Moreover, the selected flour ratio assured that the intense smell of the corn flour did not prevail, as assessed in preliminary trials.

As for the nutritional characteristics, rice and corn flours were richer in carbohydrates and presented lower contents of proteins and fibers compared to lentil flours (both NL and ECL). In particular, rice flour showed the lowest protein and fiber content. Lentil flour, instead, showed a relevant protein content, in agreement with the range of values reported in other studies [45,46]. The lentil, indeed, is known to be a protein crop, as are other pulses. In addition, the lentil is rich in lysine but poor in sulfur amino acids and is complemented by cereals such as rice and corn, which are rich in sulfur amino acids but are limited in lysine. As such, it is a practical and convenient strategy to improve the protein quality of plant-based food [47]. Therefore, NL and ECL flours were used to partly substitute rice flour at a level of 10 g/100 g of dough. Significant differences in the nutritional compositions were observed between NL and ECL, with the latter showing lower values of proteins and fibers. The difference between NL and ECL was due to thermo-mechanical degradations related to the extrusion-cooking treatment [21,26].

The gelation properties (viscoamylograph data) of the flours used for preparing the experimental GF pizza samples are reported in Table 3.

The ECL flour showed a significantly higher initial viscosity than the other flours. This result was expected, because the extrusion-cooking treatment was specifically aimed at gelatinizing the starch granules, while in the raw flours (rice, corn, and NL) the starch remained in its native form. The ECL flour showed the lowest peak viscosity, again indicating that the starch had already been completely pregelatinized during the extrusion-cooking process. The ECL flour also showed a very limited retrogradation, with significantly lower values of the cooling maximum viscosity and the setback compared to the other flours. A lower tendency towards retrogradation is appreciated in the production of bakery products, which should stay soft as long as possible. A limited setback, but greater than in ECL, was also observed in NL, confirming the results of [48]. Overall, with the exception of the initial viscosity value of ECL, all the viscoamylograph indices were lower in lentil flour than in the cereal flours, mainly due to the significantly higher protein content of the former. Proteins, indeed, tend to act as a physical barrier for starch swelling. A negative correlation between protein content with the peak viscosity was found by other authors [49,50]. The very limited breakdown of NL and ECL indicated that the paste viscosity of the lentil gels, despite the fact that they were not very elevated, were stable through the analysis and had a good shearing resistance, as already reported by other authors [51,52]. Overall, these results showed that lentil flours, and particularly ECL, were suitable for the preparation of GF pizza.

The gels from rice and corn flours showed remarkably high peak viscosity values, indicating a rapid and pronounced swelling of the native starch granules. The viscosity of gel from corn flour also remained high at its cooking temperature and under prolonged stirring (95 °C for 30 min), recording a very high value of minimum viscosity and a limited breakdown. During its subsequent cooling, the viscosity of both corn and rice gels noticeably increased, thus indicating a strong tendency towards starch retrogradation, which could be useful in noodle preparation [53] but is undesired in baked goods. These findings further justified the choice of partly replacing rice and corn flour in the preparation of GF pizza.

Regarding the better evaluation of its suitability to the preparation of GF pizza, ECL flour was also analyzed by Mixolab, in comparison with NL flour (Figure 2). Mixolab, indeed, is particularly suitable for analyzing the properties of starch associated with thermal processes, and the variations of dough consistency during mixing, simultaneously. The latter are important for the preparation of baked goods. Mixolab, therefore, substantially substitutes the separate use of the viscoamylograph and the farinograph [54]. The first part of the Mixolab curve records the dough behavior during mixing. In the second part of the analysis, the dough behavior, during its simultaneous mechanical shear stress and the temperature increase, is recorded. When a conventional gluten-containing dough is analyzed, the torque increases during the first stage until it reaches a maximum (C1). In the second part of the analysis, a decrease in the torque is usually observed, until it reaches a minimum value (C2).

The Mixolab profiles of the examined flours agreed with the viscoamylograph findings. ECL, in fact, showed a very high maximum torque in the first part of the analysis (C1). This result was due to the presence of pregelatinized starch, which increased the water absorption, producing a more consistent system that was already at 50 °C. On the contrary, NL showed an extremely low initial value of torque, because the contained starch, still native, could not gelatinize at 50 °C. Even at its minimum (C2), ECL showed a higher torque value than NL, indicating its ability to form a sufficiently structured gel to simulate a conventional (gluten-containing) dough. This ability, instead, was absent in NL which, before raising the temperature to 90 °C, never reached 0.4 Nm, which is the torque value below which the dough is considered not workable [55]. The torque of NL dough increased only with its subsequent heating at 90 °C, when the native starch of this flour gelatinized.

Viscoamylograph and Mixolab data clearly showed that rice, corn, and NL flours were not able to produce a viscous system after simple hydration, i.e., without heating at a sufficiently high temperature to achieve starch gelatinization (approximatively 95 °C). Therefore, hydrocolloids, namely HPMC (E464) and psyllium seed husk powder, were included in the formulation of pizza, to mime gluten and improve the dough structure. HPMC is known for its strong gelation capacity, which is able to increase dough viscosity [56]. This additive was not used in pizza with NL to verify the effectiveness of ECL as hydrocolloid. Psyllium husks are generally used in combination with other hydrocolloids, such as HPMC, to improve the rheological properties of GF doughs [56,57,58,59,60].

Psyllium has water-absorbing and gel-forming properties at room temperature. It is composed of arabinoxylans, whose numerous hydroxyl groups increase the capacity to bind to water and generate viscous solutions [61]. While, usually, GF products rely on the preparation of a batter, arabinoxylans act as a structuring agent that helps to retain the shape of a consistent dough without the need of a pan [62]. The structuring properties of psyllium were important in this research because, regarding the aim of keeping high the sensory properties of GF pizza, it was needed to obtain a sheetable dough instead of a batter.

Psyllium is also an effective anti-staling agent, appreciated in GF formulations [63]. Moreover, being a soluble fiber, it shows some health benefits, such as gut regulation, as well as blood glucose and cholesterol control [64]. Therefore, psyllium was also included in the formulation of GF pizza to elevate the fiber content of the final products.

### 3.2. Characteristics of GF Pizza Crust

Table 4 reports the nutritional characteristics of the experimental GF pizza samples. The addition of lentil flour had a positive impact on the nutritional profile of pizza, determining a significant increase in protein and fiber content.

In particular, the NL and ECL pizza samples were able to provide >12% of the energy value by adding proteins and could, therefore, be labeled with the nutritional claim “a source of proteins” according to the Regulation (EC) No. 1924/06 [65]. Moreover, all pizza types were able to provide more than 3 g/100 g of fibers, which could also add the “source of fibers” claim [65]. The control pizza reached the level of fiber, due to the presence of psyllium.

The raffinose family, oligosaccharides (RFOs), or α-galactosides, were also determined (Table 5), because these undigestible oligosaccharides are usually found in legumes, including lentils [66]. RFOs include raffinose, verbascose, and stachyose, and are mostly considered as antinutrients, causing flatulence and intestinal discomfort [67]. At the same time, since RFOs act as substrate for intestinal bacteria, they are also considered as prebiotics [68]. These compounds were not detected in the control pizza. This result was expected because the content of RFOs in rice and corn is generally quite low [69]. Moreover, the detrimental effect of baking, and the dilution with other ingredients, further lowered their amount. Instead, RFOs were detected both in NL and ECL pizzas. With the exception of stachyose, which was the most abundant oligosaccharide—as observed in previous works [66]—no significant differences were observed between the NL and ECL pizzas. The amount of stachyose, instead, was minor in the ECL compared to the NL, pizza, due to the thermal effect known to reduce these compounds. Other authors, in fact, observed a decrease in lentil oligosaccharides with cooking (boiling or roasting) [70].

As for bioactive compounds (Table 6), high levels of anthocyanins were assessed in the NL pizza. Anthocyanins, indeed, are known to be abundant in the red lentil [71]. Lower amounts of anthocyanins were observed in the ECL, compared to the NL, pizza, due to the thermal degradation [72] that occurred during extrusion-cooking. A relevant reduction in the level of total anthocyanins has also been observed during the extrusion-cooking of blue and red corn [73]. On the contrary, carotenoids were not affected by the extrusion-cooking process, as shown by the lack of significant differences in the carotenoid content between the NL and ECL pizzas. Similarly, other authors reported that the total carotenoid content did not decrease during the extrusion-cooking of corn [74]. Carotenoids, indeed, are known to be more heat-stable than other total anthocyanins. An average decrease in the total carotenoids of about 15% was observed after heating pigment-rich fruits, such as papaya and pineapple, at 100 °C for 8 min, whereas with the same combination of time and temperature, the content of total anthocyanins decreased by about 30% [75]. Therefore, carotenoids were abundant in all samples, even in the control, where it was contributed by corn flour. Polyphenols, instead, were quantified in lower amounts. Overall, these compounds can prevent or reduce lipid oxidation and can scavenge oxygen free radicals [76], as well as having a strong Pearson’s correlation with antioxidant activity, carotenoids, and total flavonoids [77]. Therefore, the in vitro antioxidant activity was measured in the experimental GF pizza samples. Due to the higher content of bioactive compounds, the ECL and NL pizzas showed a significantly higher antioxidant activity than the control. A lower antioxidant activity was observed in the ECL pizza, compared to the NL pizza, but this difference was not statistically significant.

As for the physical characteristics of GF pizza, related to its quality, no significant difference in pliability was observed between the control and ECL pizzas. A good-quality pizza should be easily pliable and should have, at least in the central part, a soft consistency [78]. Instead, the NL pizza was significantly (*p* < 0.05) less pliable; therefore, it required a higher force to be bent (Table 7). This result is very interesting, as it shows that the use of ECL flour made it possible to obtain a baked product with good pliability even without the addition of HPMC, while the NL pizza was unable to achieve the mechanical characteristics of the control in the absence of HPMC.

The color of baked goods, including pizza, is due to the combined effect of the Maillard reaction, induced by baking, and the contribution of bioactive pigments, such as anthocyanins and carotenoids, that are present in the flours. In fact, the color of the experimental GF pizza samples was influenced by the addition of both lentil flours, with a significant (*p* < 0.05) decrease in L* and an increase in a* and b*, compared with the control pizza. This result was due to the color of the lentil flour used: the NL flour was reddish and the ECL flour was amber yellow (Figure 3). As a consequence, the NL pizza showed also a significantly higher a* value than the ECL pizza. Color is a fundamental component of food appearance, known to influence the willingness to buy, as expressed by consumers [79]. Color alterations have already been observed as a consequence of the addition of pulse flour to pizza crusts [17], depending on the color of the pulse flour added. In view of its reduction, it is advisable to carry out a preliminary selection of the fortifying flours by preferring those flours which, with the addition of water, give the dough a similar appearance to the traditional wheat dough.

As for the total color difference, compared to the control (∆E), the ECL pizza showed a perceptible difference in color, but this difference was even greater between the control pizza and the NL pizza, as can be observed in Figure 4.

Baking involves physical changes in the bakery products, related to the increase in volume which, in turn, is due to the heat-induced expansion of the gases incorporated into the dough during the kneading and fermentation phases. This increase in volume resulted in an associated increase in the thickness of pizza samples (up to 111.1%), and a moderate decrease in diameter (10–13%). Similar values were observed in pizza prepared from blends of wheat and chickpea flour [17]. The increase in thickness demonstrated a good leavening degree and, above all, showed that a sufficiently developed and extensible dough was obtained, which was able to retain the gases. The highest increase in thickness was observed in the control pizza, while the lowest was observed in the NL pizza; however, the differences among the samples were not statistically significant.

As for the sensorial profile (Table 8), the addition of ECL and NL intensified (but not significantly) the global odor of pizza compared to the control, mostly by contributing a moderate lentil odor. The latter was perceived without any significant differences between the ECL and NL pizzas. Moreover, the color was perceived as significantly different among the three pizza types, with the NL pizza being the most intensely colored, showing orange undertones. The sensory evaluation of color agreed with the colorimeter measures.

The control pizza and the ECL pizza did not show significant differences in consistency, being both moderately soft, whereas the NL pizza was significantly harder and less compressible. NL pizza was also less pliable and more rigid than the control and ECL pizzas, which appeared more elastic and pliable. The control and ECL pizzas did not show significant differences between them, again in agreement with the instrumental determinations, highlighting the positive effect of ECL flour on the mechanical properties of pizza.

Flavor notes, namely, saltiness and sweetness, instead, were not influenced by the addition of lentil flours. The lentil flavor was moderately perceived in both the NL and ECL pizzas, without significant differences between them. Therefore, in an overall sensory comparison with the control, the ECL pizza differed substantially only in the intensity of the lentil odor and flavor, and in the color.

The consumer test indicated that no one expressed a dislike towards the three GF pizza types, highlighting that even the qualitative level of the NL pizza, despite being lower than the control and ECL pizzas, was not too low. The three types of pizza, therefore, were liked by all consumers, but to a different extent. The liked level of the NL pizza was lower than the control pizza, while between the ECL and control pizzas, there were no significant differences. The lower liked level of the NL pizza was probably imputable to its unusual orange color, harder consistency, and reduced pliability.

## 4. Conclusions

The obtained results showed that lentil flour can be added to GF pizza with interesting nutritional and qualitative results, particularly if extruded flours are used. In fact, the use of ECL flour, containing pregelatinized starch, made it possible to substitute HPMC, obtaining a “clean label” product. Compared to the control, the ECL pizza differed substantially only in the intensity of the lentil odor and flavor, and in color, but with no significant differences in the liked level. These findings are of great significance for people suffering from celiac disease, because the quality defects of GF pizza may contribute to an occasional lack of adherence to a strict GF diet, which is relatively frequent in adults. Moreover, a high content of bioactive compounds, coupled with a high protein content, makes the ECL pizza a potentially functional food product.

Pizza has spread all over the world and has adapted to the most diverse cultures and tastes, and it is still far from stopping in its path. The addition of extruded lentil flour is another example of its possible evolution.

## Figures and Tables

**Figure 1 foods-11-00482-f001:**
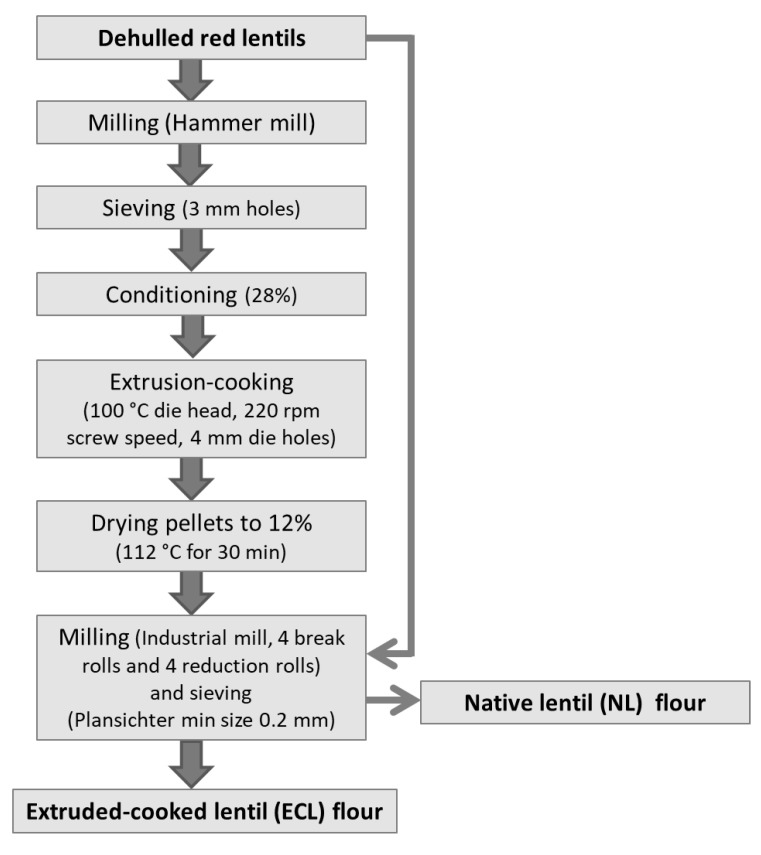
Flow chart of the productive process of native and extruded-cooked lentil flour.

**Figure 2 foods-11-00482-f002:**
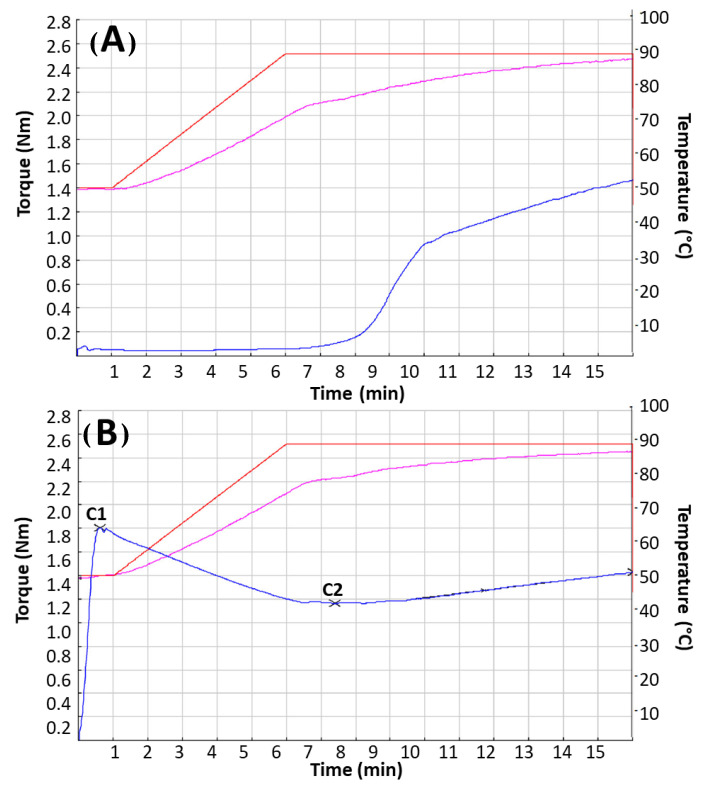
Mixolab profile of native (**A**) and extruded-cooked (**B**) lentil flour. The red line indicates the variation of Mixolab temperature setting, the violet line indicates the variations of the actual temperature of the dough, and the blue line indicates the variations of torque. C1 = maximum torque at the beginning of the analysis, C2 = minimum torque while the dough was subjected to mechanical stress.

**Figure 3 foods-11-00482-f003:**
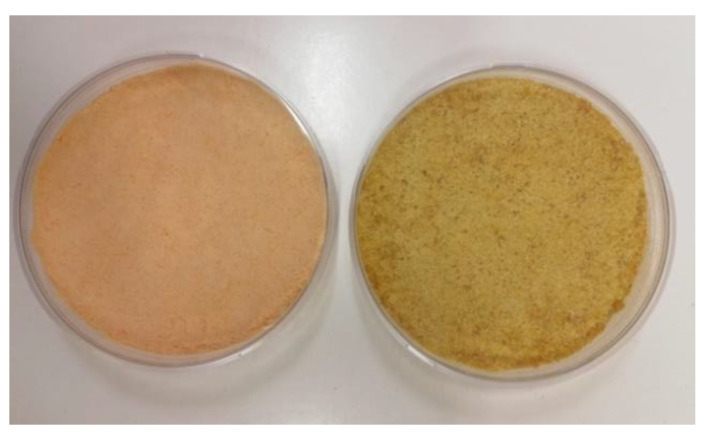
Visual appearance of native (**left**) and extruded-cooked (**right**) lentil flour.

**Figure 4 foods-11-00482-f004:**
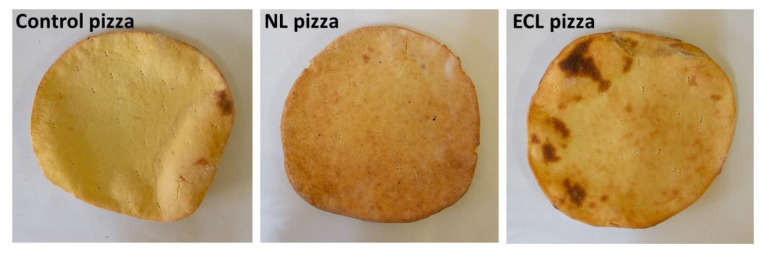
Visual appearance of experimental GF pizza samples. NL pizza = pizza crust enriched with native lentil flour; ECL pizza = pizza crust enriched with extruded-cooked lentil flour.

**Table 1 foods-11-00482-t001:** Amounts of each ingredient (g/100 g of dough) used in the formulation of the experimental samples of pizza crust, where either native (NL) or extruded-cooked (ECL) lentil flour partly substituted the rice flour.

Ingredient	Type of Pizza		
	Control	NL	ECL
Rice flour	30	20	20
Native lentil flour	-	10	-
Extruded-cooked lentil flour	-	-	10
Corn flour	7.5	7.5	7.5
Corn starch	7.5	7.5	7.5
Psyllium seed husk powder	1.5	1.5	1.5
HPMC (E464)	1.0	-	-
Yeast	1.0	1.0	1.0
Salt	1.5	1.5	1.5
Water	50	48	53

HPMC = Hydroxypropylmethylcellulose.

**Table 2 foods-11-00482-t002:** Nutritional composition and energy value of the flours used for preparing the experimental GF pizza samples. NL = native lentil flour; ECL = extruded-cooked lentil flour.

Parameter	Type of Flour			
	Rice	Corn	NL	ECL
Moisture (g/100 g)	12.7 ± 0.1 ^a^	12.6 ± 0.1 ^a^	10.3 ± 0.1 ^c^	11.2 ± 0.2 ^b^
Carbohydrates (g/100 g)	76.8 ± 0.3 ^a^	74.7 ± 0.2 ^b^	52.3 ± 0.1 ^d^	53.9 ± 0.2 ^c^
Proteins (g/100 g)	7.6 ± 0.1 ^d^	8.2 ± 0.2 ^c^	29.1 ± 0.3 ^a^	28.2 ± 0.3 ^b^
Lipids (g/100 g)	0.5 ± 0.1 ^b^	1.4 ± 0.2 ^a^	1.2 ± 0.1 ^a^	1.1 ± 0.1 ^a^
Fiber (g/100 g)	2.4 ± 0.1 ^d^	3.1 ± 0.2 ^c^	7.1 ± 0.3 ^a^	5.6 ± 0.1 ^b^
Energy value (kcal/100 g)	347 ± 6 ^a^	350 ± 5 ^a^	351 ± 8 ^a^	350 + 4 ^a^

Different letters in the rows indicate significant differences (*p* < 0.05) among flours.

**Table 3 foods-11-00482-t003:** Gelation properties (viscosity values, determined by a viscoamylograph), of the flours used for preparing the experimental GF pizza samples. NL = native lentil flour; ECL = extruded-cooked lentil flour.

Parameter	Type of Flour			
	Rice	Corn	NL	ECL
Initial viscosity (BU)	28.5 ± 3.5 ^b^	35.0 ± 2.0 ^b^	25.5 ± 1.5 ^b^	69.3 ± 4.1 ^a^
Peak viscosity (BU)	1883 ± 88 ^a^	1195 ± 45 ^b^	283.5 ± 8.5 ^c^	137.2 ± 7.6 ^d^
Minimum viscosity (BU)	986.1 ± 84.3 ^a^	1022 ± 34 ^a^	275.5 ± 9.5 ^b^	121.1 ± 6.1 ^c^
Cooling maximum viscosity (BU)	2361 ± 79 ^a^	2448 ± 26 ^a^	433.5 ± 8.5 ^b^	201.3 ± 6.4 ^c^
Breakdown (BU)	896.5 ± 61.3 ^a^	173.0 ± 7.0 ^b^	13.1 ± 0.1 ^c^	16.1 ± 2.6 ^c^
Setback (BU)	1375 ± 74 ^a^	1425 ± 11 ^a^	158.1 ± 7.1 ^b^	80.3 ± 0.6 ^c^

BU = Brabender units. Different letters in the rows indicate significant differences (*p* < 0.05) among flours.

**Table 4 foods-11-00482-t004:** Nutritional composition and energy value of the experimental GF pizza samples. NL = native lentil flour; ECL = extruded-cooked lentil flour.

Parameter	Type of Pizza		
	Control	NL	ECL
Moisture (g/100 g)	38.1 ± 0.5 ^a^	39.3 ± 0.6 ^a^	38.9 ± 0.5 ^a^
Carbohydrates (g/100 g)	54.1 ± 0.8 ^a^	49.1 ± 0.2 ^b^	49.7 ± 0.7 ^b^
Proteins (g/100 g)	4.4 ± 0.1 ^b^	7.4 ± 0.1 ^a^	7.3 ± 0.2 ^a^
Lipids (g/100 g)	0.1 ± 0.1 ^a^	0.3 ± 0.2 ^a^	0.3 ± 0.1 ^a^
Fiber (g/100 g)	3.3 ± 0.1 ^b^	3.9 ± 0.2 ^a^	3.8 ± 0.3 ^ab^
Energy value (kcal/100 g)	242 ± 3 ^a^	237 ± 4 ^a^	238 ± 3 ^a^

Different letters in the rows indicate significant differences (*p* < 0.05) among samples.

**Table 5 foods-11-00482-t005:** Content of oligosaccharides in the experimental GF pizza samples. NL = native lentil flour; ECL = extruded-cooked lentil flour.

Oligosaccharide	Type of Pizza		
	Control	NL	ECL
Verbascose (mg/g d.m.)	n.d.	0.92 ± 0.04 ^a^	0.98 ± 0.03 ^a^
Stachyose (mg/g d.m.)	n.d.	5.55 ± 0.34 ^a^	4.04 ± 0.18 ^b^
Raffinose (mg/g d.m.)	n.d.	2.05 ± 0.50 ^a^	1.98 ± 0.20 ^a^

Different letters in the rows indicate significant differences (*p* < 0.05) among samples; n.d. = not detected.

**Table 6 foods-11-00482-t006:** Level of bioactive compounds and antioxidant activity of the experimental GF pizza samples. NL = native lentil flour; ECL = extruded-cooked lentil flour.

Parameter	Type of Pizza		
	Control	NL	ECL
Total anthocyanins (mg/kg cyanidin 3-*O*-glucoside d.m.)	n.d.	16.29 ± 0.05 ^a^	4.36 ± 0.17 ^b^
Total carotenoids (mg/kg β-carotene d.m.)	3.99 ± 0.08 ^b^	5.36 ± 0.26 ^a^	5.66 ± 0.11 ^a^
Total phenolic compounds (mg/g ferulic acid d.m.)	0.67 ± 0.08 ^a^	0.90 ± 0.21 ^a^	0.74 ± 0.01 ^a^
Antioxidant activity (µmol Trolox/g d.m.)	1.07 ± 0.09 ^b^	1.42 ± 0.06 ^a^	1.35 ± 0.04 ^a^

Different letters in the rows indicate significant differences (*p* < 0.05) among samples; n.d. = not detected.

**Table 7 foods-11-00482-t007:** Pliability, color, and dimensional variations of the experimental GF pizza samples. NL = native lentil flour; ECL = extruded-cooked lentil flour.

Parameter		Type of Pizza	
	Control	NL	ECL
Pliability (bending test)			
F max (N/mm^2^)	5.27 ± 1.03 ^b^	7.96 ± 1.07 ^a^	4.47 ± 1.33 ^b^
Color			
*L**	81.31 ± 0.81 ^a^	58.18 ± 0.38 ^c^	74.17 ± 0.56 ^b^
*a**	2.60 ± 0.43 ^c^	18.51 ± 0.38 ^a^	7.06 ± 0.52 ^b^
*b**	20.38 ± 0.87 ^a^	35.68 ± 1.50 ^a^	27.25 ± 0.53 ^b^
ΔE	-	31.98 ± 1.76 ^a^	11.44 ± 1.34 ^b^
Dimensional variations induced by baking			
Diameter variation (%)	−11.9 ± 1.2 ^a^	−13.2 ± 1.6 ^a^	−10.5 ± 1.8 ^a^
Thickness variation (%)	111.1 ± 9.2 ^a^	98.1 ± 4.1 ^a^	104.7 ± 5.3 ^a^

Different letters in the rows indicate significant differences (*p* < 0.05) among samples.

**Table 8 foods-11-00482-t008:** Sensory scores recorded by quantitative descriptive sensory analysis and by consumer test carried out on the experimental GF pizza samples. NL pizza = pizza crust enriched with native lentil flour; ECL pizza = pizza crust enriched with extruded-cooked lentil flour. The range of the scale was 0–9 contractual units (c.u.) for the quantitative descriptive sensory analysis and 1–5 c.u. for the consumer test.

Parameter	Type of Pizza		
	Control	NL	ECL
Quantitative descriptive sensory analysis			
Global odor	4.7 ± 0.4 ^a^	5.5 ± 0.1 ^a^	5.4 ± 0.1 ^a^
Lentil odor	0.2 ± 0.1 ^b^	3.2 ± 0.5 ^a^	3.3 ± 0.5 ^a^
Color	3.7 ± 0.2 ^c^	6.9 ± 0.6 ^a^	4.5 ± 0.2 ^b^
Consistency	5.4 ± 0.3 ^b^	6.3 ± 0.2 ^a^	5.5 ± 0.3 ^b^
Pliability	5.6 ± 0.4 ^a^	4.5 ± 0.3 ^b^	6.2 ± 0.1 ^a^
Sweetness	2.8 ± 0.3 ^a^	2.5 ± 0.4 ^a^	2.6 ± 0.3 ^a^
Saltiness	1.7 ± 0.2 ^a^	1.9 ± 0.3 ^a^	2.1 ± 0.4 ^a^
Lentil flavor	0.1 ± 0.1 ^b^	4.6 ± 0.4 ^a^	4.3 ± 0.1 ^a^
Consumer test			
Dislike *	0.0 ± 0.0 ^a^	0.0 ± 0.0 ^a^	0.0 ± 0.0 ^a^
Like *	3.8 ± 1.2 ^a^	3.3 ± 1.3 ^b^	3.4 ± 1.2 ^ab^

Different letters in the rows indicate significant differences (*p* < 0.05) among samples. * Dislike and like data from the consumer test were submitted to the Friedman non-parametric statistical test.

## Data Availability

The data presented in this study are available upon request from the corresponding author.

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
