# Peer review of "The Effectiveness of Extruded-Cooked Lentil Flour in Preparing a Gluten-Free Pizza with Improved Nutritional Features and a Good Sensory Quality"

_foods, 2022, doi:10.3390/foods11030482_

Round 1

Reviewer 1 Report

In the manuscript titled "Effectiveness of extruded-cooked lentil flour in preparing gluten-free pizza with improved nutritional features and good sensory quality" characteristic of extruded-cooked lentil flour is analysed as well as its possible addition to gluten-free pizza.

The research of the production or ingredients of gluten-free food is crucial to many people, especially suffering from celiac disease.

The article is written very well .

I have one general suggestion that in the conclusion could be underlineg the significance of the research for people suffering from celiac disease.

A few detailed comments:

Line 122: Table 1, there is no unit of mass of ingredients.

Line 192: Please, add information regarding the scale (structured/unstructured) and the units - contractual units [c.u]

Line 366: I suggest adding the symbols of the pizzas on the pictures.

Line 389: Flavor, no taste.

Line 401: Please, add to the caption range of the scale 0-9 contractual units [c.u.]

Author Response

Reviewer 1

In the manuscript titled "Effectiveness of extruded-cooked lentil flour in preparing gluten-free pizza with improved nutritional features and good sensory quality" characteristic of extruded-cooked lentil flour is analysed as well as its possible addition to gluten-free pizza.

The research of the production or ingredients of gluten-free food is crucial to many people, especially suffering from celiac disease.

The article is written very well and meets the requirements of a research paper as well as the Foods.

Response: We thank the reviewer for his/her positive evaluation of our work.

I have one general suggestion that in the conclusion could be underlined the significance of the research for people suffering from celiac disease.

Response: Thank for this suggestion. A sentence regarding the significance of the research for people suffering from celiac disease has been added to the conclusions.

A few detailed comments:

Line 122: Table 1, there is no unit of mass of ingredients.

Response: Sorry for the mistake, we amended it by adding the unit.

Line 192: Please, add information regarding the scale (structured/unstructured) and the units - contractual units [c.u]

Response: The scale was structured. We added this specification, as well as the contractual units, in the text.

Line 366: I suggest adding the symbols of the pizzas on the pictures.

Response: The acronyms of the pizza samples have been added in the pictures.

Line 389: Flavor, no taste.

Response: Modified as suggested.

Line 401: Please, add to the caption range of the scale 0-9 contractual units [c.u.]

Response: Thanks, we added the suggested specification to the Table caption.

Reviewer 2 Report

I think that the manuscript entitled “Effectiveness of extruded-cooked lentil flour in preparing gluten-free pizza with improved nutritional features and good sensory quality" deserves publication in the Foods after major revision. The paper presents a number of different research methods, however, on a small number of research facilities (3). The manuscript should be supplemented with a thorough discussion of the results obtained with the results of other authors researching similar problems.

Table 1: components which unit

Line 126-127: fat analysis - literature

Line 124: please describe the methods used, broken down by constituents, basic and bioactive ingredients, with the reagents and equipment used.

 Line 160: please change „naximum” into „ maximum”

Line 176: please complete in the information on the conditions for conducting a professional sensory analysis

Line 192: why is the scale 0-9 and not 0-10 as in professional sensory analysis?

Line 209: please change „p<0.05” into „ p < 0.05”

Table 2: please change „kcal” into „kcal/100g”

Figure 2: please complete what do the lines mean

All Table what the values in the tables mean

Figure captions are on a different pages then the figures

To write a discussion of the results which practically does not exist and is a very important part of the manuscript.

Author Response

Reviewer 2

I think that the manuscript entitled “Effectiveness of extruded-cooked lentil flour in preparing gluten-free pizza with improved nutritional features and good sensory quality" deserves publication in the Foods after major revision. The paper presents a number of different research methods, however, on a small number of research facilities (3). The manuscript should be supplemented with a thorough discussion of the results obtained with the results of other authors researching similar problems.

Response: Thanks for your observation. We implemented the discussion with additional comments especially in comparison with current literature. In detail, 9 new references have been considered and added.

Table 1: components which unit

Response: Sorry for the mistake, we amended it by adding the unit.

Line 126-127: fat analysis - literature

Response: A reference has been added for fat analysis.

Line 124: please describe the methods used, broken down by constituents, basic and bioactive ingredients, with the reagents and equipment used.

Response: For the entire section “Chemical Determinations”, a more detailed description of the methods used, broken down by constituents, basic and bioactive ingredients, with the reagents and equipment used has been added.

 Line 160: please change „naximum” into „ maximum”

Response: We amended the typo, thanks for noting.

Line 176: please complete in the information on the conditions for conducting a professional sensory analysis

Response: Information on the conditions for conducting a professional sensory analysis have been added.

Line 192: why is the scale 0-9 and not 0-10 as in professional sensory analysis?

Response: Thanks for your observation, the scale was 0-9 to have a 10-point evaluation (including 0).

Please consider that our main reference, since when we had our first collaboration with her, in 2003, are the works of Dr. Fiorella Sinesio, who has many years of experience in sensory evaluation at the CREA – Research Centre for Food and Nutrition, Rome, Italy, where leads a panel of professional sensory assessor for the evaluation of many types of food, including baked goods (see, for example: Sinesio, F., Raffo, A., Peparaio, M., Moneta, E., Civitelli, E.S., Narducci, V., Turfani, V., Nicoli, S.F. and Carcea, M., 2019. Impact of sodium reduction strategies on volatile compounds, sensory properties and consumer perception in commercial wheat bread. Food Chemistry, 301, p.125252; Raffo, A., Carcea, M., Moneta, E., Narducci, V., Nicoli, S., Peparaio, M., Sinesio, F. and Turfani, V., 2018. Influence of different levels of sodium chloride and of a reduced-sodium salt substitute on volatiles formation and sensory quality of wheat bread. Journal of Cereal Science, 79, 518-526; Raffo, A., Pasqualone, A., Sinesio, F., Paoletti, F., Quaglia, G. and Simeone, R., 2003. Influence of durum wheat cultivar on the sensory profile and staling rate of Altamura bread. European Food Research and Technology, 218, 49-55).

Also other research groups adopt a 0-9 scale, see for example: Vilanova, M., 2006. Sensory descriptive analysis and consumer acceptability of godello wines from Valdeorras apellation origen controlée (Northwest Spain). Journal of sensory studies, 21, 362-372; De Pelsmaeker, S., Gellynck, X., Delbaere, C., Declercq, N. and Dewettinck, K., 2015. Consumer-driven product development and improvement combined with sensory analysis: A case-study for European filled chocolates. Food Quality and Preference, 41, 20-29; Cozzolino, D., Smyth, H.E., Lattey, K.A., Cynkar, W., Janik, L., Dambergs, R.G., Francis, I.L. and Gishen, M., 2005. Relationship between sensory analysis and near infrared spectroscopy in Australian Riesling and Chardonnay wines. Analytica Chimica Acta, 539, 341-348.

Actually, we are aware that there are also many other research papers where a 0-10 scale was adopted, but as I mentioned we preferred to have a 10-point evaluation, not an 11-point one. Anyway, by specifying the scale the reader can put results in context and understand the extent of each evaluation.

Line 209: please change „p<0.05” into „ p < 0.05”

Response: Amended as suggested.

Table 2: please change „kcal” into „kcal/100g”

Response: Thanks, we amended the mistake (also in Table 4).

Figure 2: please complete what do the lines mean

Response: Thanks, we completed the caption of Figure 2 specifying the meaning of all the three lines.

All Table what the values in the tables mean

Response: In all tables it has been specified what the values mean.

Figure captions are on a different pages then the figures

Response: Figure captions have been checked and kept on the same pages then the figures

To write a discussion of the results which practically does not exist and is a very important part of the manuscript.

Response: Sorry that you found the discussion not sufficient; we implemented it with additional comments especially in comparison with current literature. In detail, 9 new references have been considered and added.

Reviewer 3 Report

      In my opinion, the manuscript entitled  ,,Effectiveness of extruded-cooked lentil flour in preparing gluten-free pizza with improved nutritional features and good sensory quality’’ by Pasqualone et al., aimed to replace the conventional flours used for gluten free pizza manufacturing with extruded-cooked lentil flour. The introduction is according with the current state of the art, the methods are summary described but enough in order to be reproduced, the results are well discussed and compared with other studies.

I only have some small comments as follows:

Line 19. Please mentioned according to which rules or regulations the new obtained products could be named ,,source of proteins’’? or deleted according to current rules.

Line 2.1. Basic ingredients

Please also mention the source for the reagents used in order to analyzed protein, total phenolic compounds, total carotenoids, total dietary fiber and antioxidant activity.

Line 85: please replace at with from; were purchased from local retailers, not at local retailers.

Lines 95-96. How did you choose the working parameters for extruder-cooker?

Which was the compression ratio of the extruder and the dosing speed (feed rate range?). Please complete the text with the required information.

Line 114. Please mention the rpm for the mixed ingredients at 2 minutes and 10 min, respectively.

Line 335. How can you explain that total carotenoids were not so affected by the extrusion process compared to the total anthocyanins? For instance, in a recent study, Igual et al 2021, entitled Effect on Nutritional and Functional Characteristics by Encapsulating Rose canina Powder in Enriched Corn Extrudates the authors identified a strong Pearson correlation between antioxidant activity, carotenoids, total flavonoids and vitamin C.

Please introduced in the present manuscript a short phrase to better describe the carotenoids content of the obtained pizza types.

Thank you!

Author Response

Reviewer 3

In my opinion, the manuscript entitled ,,Effectiveness of extruded-cooked lentil flour in preparing gluten-free pizza with improved nutritional features and good sensory quality’’ by Pasqualone et al., aimed to replace the conventional flours used for gluten free pizza manufacturing with extruded-cooked lentil flour. The introduction is according with the current state of the art, the methods are summary described but enough in order to be reproduced, the results are well discussed and compared with other studies.

Response: We thank the Reviewer for positively evaluating our work.

I only have some small comments as follows:

Line 19. Please mentioned according to which rules or regulations the new obtained products could be named ,,source of proteins’’? or deleted according to current rules.

Response: The specific rule (Regulation (EC) No 1924/2006) has been mentioned in the text (see line 19).

Line 2.1. Basic ingredients

Response: We modified as suggested (see line 81).

Please also mention the source for the reagents used in order to analyzed protein, total phenolic compounds, total carotenoids, total dietary fiber and antioxidant activity.

Response: The source for the reagents used in order to analyzed protein, total phenolic compounds, total carotenoids, total dietary fiber and antioxidant activity has been specified (see lines 132-134; 141-144; 167-168; 172-173; 177-178).

Line 85: please replace at with from; were purchased from local retailers, not at local retailers.

Response: Thanks, we amended it (see line 85).

Lines 95-96. How did you choose the working parameters for extruder-cooker?

Response: The extrusion-cooking conditions were selected after preliminary tests because they allowed to obtain an extruded flour with acceptably high cold viscosity (69.3 Brabender Units - BU). This specification has been added in the text (see lines 97-100).

Which was the compression ratio of the extruder and the dosing speed (feed rate range?). Please complete the text with the required information.

Response: The text has been completed with the required information (see lines 96-97).

Line 114. Please mention the rpm for the mixed ingredients at 2 minutes and 10 min, respectively.

Response: The text has been completed with the rpm of the kneader during the 2 min and the subsequent 10 min mixing steps (see line 117 and 120).

Line 335. How can you explain that total carotenoids were not so affected by the extrusion process compared to the total anthocyanins? For instance, in a recent study, Igual et al 2021, entitled Effect on Nutritional and Functional Characteristics by Encapsulating Rose canina Powder in Enriched Corn Extrudates the authors identified a strong Pearson correlation between antioxidant activity, carotenoids, total flavonoids and vitamin C.

Response: Thanks for comment. The greater heat-stability of carotenoids, compared to anthocyanins, has been discussed with references and the strong correlation with antioxidant activity has been highlighted based on the suggested study (see lines 421-430 and lines 433-434).

Please introduced in the present manuscript a short phrase to better describe the carotenoids content of the obtained pizza types.

Response: A short phrase to describe the carotenoid content of the obtained pizza types has been added (see lines 424-425).

Round 2

Reviewer 2 Report

I think that the re-submitted the manuscript entitled “Effectiveness of extruded-cooked lentil flour in preparing gluten-free pizza with improved nutritional features and good sensory quality" deserves publication in the Foods in present form.

Please read ISO 13299:2016 and ISO 11132:2021 when writing subsequent publications using sensory profiling analysis.

Author Response

Reviewer 2

I think that the re-submitted the manuscript entitled “Effectiveness of extruded-cooked lentil flour in preparing gluten-free pizza with improved nutritional features and good sensory quality" deserves publication in the Foods in present form.

Response: We thank the reviewer for his/her positive evaluation of our work.

Please read ISO 13299:2016 and ISO 11132:2021 when writing subsequent publications using sensory profiling analysis.

Response: Former reference 41 (now 42, which was moved in another point of the text) was a manuscript from our research group specifically regarding the variations of the sensory profile of a bakery product (namely Altamura PDO (Protected Designation of Origin) bread), where we wrote that “Quantitative descriptive analysis of the sensory properties was carried out according to standard 13299 of the Intl. Organization for Standardization (ISO)”. That is why we did not repeat mentioning the 13299 ISO Standard. Considering your suggestion, for completeness, in this R2 revision we modified the sentence explicitly mentioning the 13299 ISO Standard. Thanks for suggesting it.

Regarding the guidelines given by ISO 11132:2021 to measure the performance of the quantitative descriptive sensory panel, we already wrote in the manuscript that we verified the reliability, consistency, and discriminating ability of panelists. Since the way we verified these requirements was the same way as described in the ISO standard, we cited it in this revised version. Thanks again for suggesting it.